# Muscle and Bone Impairment in Infantile Nephropathic Cystinosis: New Concepts

**DOI:** 10.3390/cells11010170

**Published:** 2022-01-05

**Authors:** Dieter Haffner, Maren Leifheit-Nestler, Candide Alioli, Justine Bacchetta

**Affiliations:** 1Department of Pediatric Kidney, Liver and Metabolic Diseases, Hannover Medical School, Carl-Neuberg-Str. 1, 30625 Hannover, Germany; Leifheit-Nestler.Maren@mh-hannover.de; 2Pediatric Research Center, Hannover Medical School, Carl-Neuberg-Str. 1, 30625 Hannover, Germany; 3INSERM Research Unit 1033, Pathophysiology of Bone Disease, Faculté de Médecine Lyon Est, Université de Lyon, Rue Guillaume Paradin, 69008 Lyon, France; candide.alioli@inserm.fr (C.A.); justine.bacchetta@chu-lyon.fr (J.B.); 4Reference Center for Rare Renal Diseases, Reference Center for Rare Diseases of Calcium and Phosphate Metabolism, Pediatric Nephrology, Rheumatology and Dermatology Unit, Hôpital Femme Mère Enfant, Boulevard Pinel, 69500 Bron, France

**Keywords:** infantile nephropathic cystinosis, bone-muscle wasting, fibroblast growth factor 23, osteoclasts, sclerostin, leptin, fractures, cysteamine

## Abstract

Cystinosis Metabolic Bone Disease (CMBD) has emerged during the last decade as a well-recognized, long-term complication in patients suffering from infantile nephropathic cystinosis (INC), resulting in significant morbidity and impaired quality of life in teenagers and adults with INC. Its underlying pathophysiology is complex and multifactorial, associating complementary, albeit distinct entities, in addition to ordinary mineral and bone disorders observed in other types of chronic kidney disease. Amongst these long-term consequences are renal Fanconi syndrome, hypophosphatemic rickets, malnutrition, hormonal abnormalities, muscular impairment, and intrinsic cellular bone defects in bone cells, due to *CTNS* mutations. Recent research data in the field have demonstrated abnormal mineral regulation, intrinsic bone defects, cysteamine toxicity, muscle wasting and, likely interleukin-1-driven inflammation in the setting of CMBD. Here we summarize these new pathophysiological deregulations and discuss the crucial interplay between bone and muscle in INC. In future, vitamin D and/or biotherapies targeting the IL1β pathway may improve muscle wasting and subsequently CMBD, but this remains to be proven.

## 1. Introduction

Infantile nephropathic cystinosis (INC) is a rare autosomal recessive storage disease, due to mutations in the *CTNS* gene encoding for the lysosomal cystine transporter cystinosin [1]. CTNS malfunction results in an accumulation of cystine in all organs, primarily the kidneys, leading to Fanconi syndrome, a global defect of the proximal renal tubules, and progressive chronic kidney disease (CKD), which can be ameliorated by early treatment with the cystine-depleting agent cysteamine [2,3]. Severe bone and muscle impairment are other important complications of INC, which often persist or are even aggravated, despite measures for Fanconi syndrome, cysteamine therapy, and kidney replacement therapy (dialysis or kidney transplantation) [4,5,6,7]. Initial studies suggest that bone and muscle impairment in INC are the primary consequences of Fanconi syndrome, and later, of mineral and bone disorder associated with CKD (CKD-MBD) [8,9]. The former results in impairment of calcium and phosphate homeostasis, with the clinical consequences of hypophosphatemic rickets and muscle weakness, whereas the latter describes the complexity of renal osteodystrophy, alterations in mineral and vitamin D metabolism, and cardiovascular complications, as seen in CKD patients with other underlying causes of CKD [2,10]. Recent clinical and experimental studies provide increasing evidence that bone and muscle impairment in INC is much more complex and, at least partly, due to an intrinsic bone defect and elevated leptin signaling, promoting muscle wasting [11,12,13,14,15,16,17]. The term cystinosis metabolic bone disease (CMBD) was coined by an international guideline initiative to describe this complex bone phenotype in INC patients (Figure 1) [18]. This review highlights the recent insights in the pathophysiology of muscle and bone impairment and their interplay in INC.

## 2. Clinical Presentation of Bone Disease in Cystinosis

The early use (<2 years) of cysteamine treatment has postponed end-stage kidney disease and extra-renal complications beyond the second decade of life [19]. However, as patient survival improves [20], bone impairment was recently described as a “novel” complication of cystinosis resulting in substantial comorbidity during puberty or early adulthood [5,6,7,8,21]. This complication may dramatically alter the quality of life of patients: “As patients with cystinosis now routinely survive well into adulthood, additional challenges to lifelong bone health have emerged” [8]. In 2016, we reported on three teenagers with INC, displaying a severe bone phenotype that was characterized by unusual resorption areas on bone biopsy [5]. Later, a French observational study was carried out on 10 INC patients (median age 23 years, range 10–35) using biomarkers and High Resolution peripheral Quantitative Computed Tomography (HR-pQCT) at the ultra-distal tibia [6]. Seven patients (70%) complained of at least one bone symptom, namely, bone pain and/or deformation and/or history of fractures. Significant alterations in cortical parameters and, notably, cortical thickness were noted in INC patients compared to healthy subjects, similar to what was observed in *CTNS*^−/−^ knockout mice (*vide infra*). Circulating and urinary calcium, as well as alkaline phosphatase (ALP) levels, were normal. However, there was a tendency towards low parathyroid hormone (PTH) and low fibroblast growth factor 23 (FGF23) levels, likely reflecting the consequences of chronic tubular phosphate wasting, although Fanconi syndrome appeared to be well controlled in these patients. These results were further confirmed in 2020, both in a German and an American study, clearly showing decreased FGF23 levels in INC patients when corrected for the stage of CKD [13,14].

A North American team also performed bone and mineral evaluations in 30 INC patients with a mean age of 20 years (range 5–44 y) [7]. Bone mineral density was reduced at all sites; low bone mass in at least one evaluation zone was present in 46% of patients. A large proportion of patients displayed bone symptoms: one or more fractures of the long bones (27%), incidental vertebral fractures (32%), limb deformities (34%), and scoliosis (50%).

An international guideline on the diagnosis and management of CMBD was published in 2019 [18]. Briefly, assessment of CMBD involves; regular monitoring of longitudinal growth, blood levels of phosphate, calcium, ALP, and bicarbonate, and, depending on the clinical and biochemical findings, bone radiography, and assessment of hormone levels and vitamins, such as thyroid hormone, PTH, 25 OH vitamin D, and testosterone in males, and surveillance for non-renal complications of INC including myopathy. Patients require a multi-disciplinary and comprehensive management of CMBD. Urinary loss due to Fanconi syndrome should be replaced, including phosphate and bicarbonate). Patients require adequate cystine-depleting treatment with cysteamine, adequate caloric and protein intake, supplementation with native vitamin D, in cases of vitamin D deficiency, treatment with active vitamin D to support the treatment of rickets, and eventually hormone replacement/therapy (thyroid hormone, testosterone in males, and recombinant growth hormone in cases of persistent short stature) [22], physical therapy, and orthopedic surgery in cases of persistent, significant limb deformities despite adequate treatment for Fanconi syndrome [18].

## 3. Cystinosis Metabolic Bone Disease

The pathophysiology of the bone phenotype in INC is complex. At least eight distinct factors contribute to CMBD (see Figure 1), which are outlined in the following section [18].

### 3.1. Fanconi Syndrome

Fanconi syndrome, due to cystine accumulation in renal proximal tubules, emerges around the age of 6 months, resulting in renal phosphate wasting and consecutive hypophosphatemia, metabolic acidosis, 1,25-vitamin D deficiency, and hypocalcemia. All of these factors promote the development of hypophosphatemic rickets [2,13,18]. However, hypophosphatemia is thought to be the most crucial factor, as it was shown to impair apoptosis of hypertrophic chondrocytes in the growth plate, the decisive cellular defect in all forms of rickets [23]. In addition, metabolic acidosis was shown to impair both, bone mineralization and linear growth in children [24,25].

### 3.2. Deficiency in Nutrition and Micronutrition

Children with INC often present with impaired appetite and frequent vomiting, due to excessive fluid intake demands due to Fanconi syndrome-associated polyuria, resulting in caloric and protein malnutrition and eventually copper deficiency, which all contribute to impaired bone growth [26,27].

### 3.3. Hormonal Disturbances

Hormonal disturbances due to cystine accumulation in endocrine organs may result in hypothyroidism, hypogonadism, and hypoparathyroidism, all of which are known inhibitors of longitudinal bone growth [28,29]. Gonadal dysfunction affecting males is characterized by hypergonadotropic hypogonadism, i.e., high luteinizing hormone (LH) and follicle-stimulating hormone (FSH), in association with low testosterone [30]. This is due to the lysosomal overload of Sertoli and Leydig cells in the testes, which can be ameliorated by cysteamine therapy [31]. In addition, INC patients develop an insensitivity to the actions of growth hormone (GH) and insulin-like growth factor 1 (IGF1), as noted in other patients suffering from advanced CKD, which further hinders growth and can be overcome by treatment with recombinant human GH [22,32,33].

### 3.4. Myopathy

Distal myopathy presents around the second decade of life in children with INC and primarily involves the hand muscles [28,34]. It may even be detected in patients with no overt muscle weakness. Later, patients may develop dysphagia and restrictive lung disease [35,36,37]. Myopathy is mainly due to cystine accumulation in striated muscles, which may be aggravated by concomitant hypophosphatemia and hypocalcemia due to Fanconi syndrome. Biopsy studies of affected muscles show marked fiber-size variability, vacuoles, and impaired grouping of fiber type [38]. Interestingly, cysteine crystals were found in perimysial cells, whereas no crystals could be detected in cell vacuoles. Electrophysiological studies of affected muscles show diminished amplitude and duration. Muscle weakness may initially be mild, primarily involving intrinsic hand muscles. Later, patients show pronounced distal weakness, more so than proximal weakness and contractures [35]. A recent study in 76 pediatric and adult INC patients showed a mean grip strength SD-score of −2.1, which was significantly lower compared to CKD patients with other underlying kidney diseases [39]. Reduced grip strength was associated with the male gender, delayed cysteamine therapy, and low levels of physical activity. Impaired muscle function leads to impaired bone health through reduced mechanical loads on bone. This is supported by the findings that the strength of load-bearing bones largely depends on growing muscle strength [40].

### 3.5. Mineral and Bone Disorders Due to CKD (CKD-MBD)

CKD-MBD, due to a progressive decline in glomerular function, leads to impaired bone health, well known from CKD patients suffering from other kidney diseases [10]. The term CKD-MBD has replaced the old term renal osteodystrophy, which suggests that bone changes in CKD patients are primarily due to secondary hyperparathyroidism and vitamin D deficiency [41]. The first detectable bone abnormality in mild CKD not associated with renal Fanconi syndrome is an increased expression in the number of sclerostin and FGF23-expressing osteocytes, causing progressive elevations in the plasma concentrations of these bone-derived factors [41,42,43]. FGF23 acts, as PTH, as a phosphaturic hormone by inhibiting sodium (Na)-dependent phosphate (Pi) reabsorption via NaPi2a/2c after binding to the FGF receptor 1 and its cofactor Klotho [44]. This allows for normal phosphate homeostasis in the early stages of CKD, despite impaired renal excretory capacity. Elevated FGF23 levels also impair synthesis and increase degradation of calcitriol (1,25(OH)_2_D), thereby promoting vitamin D deficiency [45]. Sclerostin was shown to directly inhibit the Wnt/Beta-catenin pathway in osteocytes. Its synthesis is already increased in patients with CKD stage 2, resulting in reduced bone remodeling [46]. Therefore, the current concept of CKD-MBD is that both increased FGF23 and sclerostin causes early bone loss in the early stages of CKD. It was thought that this is also the case in patients with INC. Recent studies show that INC is characterized by distinct CKD stage-dependent abnormalities in bone metabolism, including sclerostin and FGF23, which differs markedly to that observed in patients with other underlying causes of CKD as outlined below [13,14].

### 3.6. CKD-MBD Post Kidney Transplantation

Persistent elevation of both FGF23 and PTH levels are often noted in patients who received a kidney transplant, despite excellent transplant function, which may promote bone disease in these patients [47]. In addition, post-transplant growth can be markedly diminished due to glucocorticoid treatment which may also promote osteoporosis and increased risk of fractures in these patients, irrespective of the underlying renal disease [48].

### 3.7. Intrinsic Bone Defect

Cystinosin is expressed in bone cells, including osteoblasts and osteoclasts. *CTNS*^−/−^ knockout mice do not develop overt Fanconi syndrome for, so far, unknown reasons [12]. Despite this, they display a clear bone phenotype, characterized by reduced trabecular bone, cortical thickness, and bone mineral density compared to healthy animals [12]. In addition, reduced osteoblast and osteoclast parameters were noted on tibiae histomorphometry of *CTNS*^−/−^ knockout mice compared to controls. In vitro experiments using osteoblasts from *CTNS*^−/−^ knockout mice demonstrated elevated cystine content, reduced numbers of ALP-positive cells, diminished expression of differentiation and activity markers, and impaired mineralization, compared to cells taken from wild-type mice. Conforti et al. also reported that mesenchymal stem cells derived from an INC patient show a diminished ability to differentiate into osteoclasts [49]. Taken together, these observations suggest an intrinsic osteoblast and osteoclast defect in cystinosis.

This was further evaluated in studies using peripheral blood mononuclear cells (PBMCs) derived from INC patients. Claramunt-Taberner et al., demonstrated that CTNS was clearly expressed in human PBMCs derived from healthy subjects [11]. In addition, PBMCs derived from INC patients showed an increased number of tartrate-resistant acid phosphatase 5b (TRAP5b)-positive cells compared to healthy subjects, suggesting that cystinosis favors osteoclastogenesis [11]. The same group performed a subsequent study using PBMCs from patients with *CTNS* variants and residual cystine efflux activity, with inactive *CTNS* variants and with *CTNS* variants not allowing proper protein translation and presentation at the lysosomal membrane [15]. Interestingly, PBMCs with residual CTNS activity generated less osteoclasts compared to those with inactive or absent CTNS, indicating that CTNS may act as a negative regulator of osteoclast formation. In other words, loss of CTNS function may cause increased osteoclast activity. This is also supported by clinical studies in INC patients, using serum TRAP5b levels as a measure of osteoclast function [13]. Lysosomal dysfunction and defective autophagic mitochondria clearance have recently been showed in epithelial tubular cells [50], inducing increased oxidative cells. Since osteoclasts are giant multinucleated cells with a very high mitochondrial density, it is tempting to hypothesize a mitochondrial defect in cystinotic bone, but this remains to be proved [4].

### 3.8. Cysteamine Toxicity

There is increasing evidence that cysteamine treatment affects osteoblast and osteoclast function. In vitro studies using PBMCs from healthy donors and INC patients showed that cysteamine does not modify osteoclastogenesis [11]. However, high doses of cysteamine resulted in a clear reduction in bone resorption by PBMCs derived from INC patients, whereas low cysteamine doses stimulated osteoblastic differentiation and mineralization. This was later confirmed in a second study by the same group, showing that high dose cysteamine treatment exerts an inhibitory effect on osteoclastic differentiation in PBMCs derived from INC patients, irrespective of the severity of *CTNS* mutation, i.e., residual CTNS activity versus inactive or absent CTNS [15]. Thus, cysteamine, if given at high doses, may also impair bone health in INC patients.

## 4. Bone and Mineral Metabolism in INC Patients

Recently, the circulating parameters of bone and mineral metabolism were investigated in a cohort of 49 European children and adolescents with INC compared to 80 patients with other CKD entities [13]. The main parameters included; FGF23, soluble Klotho (sKlotho), sclerostin, bone alkaline phosphatase (BAP)—as a marker of bone formation, TRAP5b as an osteoclast marker and the receptor activator for the nuclear factor kappa-B ligand (RANKL)/osteoprotegerin (OPG) system, which is a strong regulator of osteoclast formation and activity. As expected, normalized serum phosphate levels were clearly decreased in INC patients with mild–moderate CKD compared to healthy children and CKD-controls, despite oral treatment with phosphate salts due to persistent renal phosphate wasting (Figure 2). INC patients showed a high frequency of reduced levels of phosphate, calcium PTH, and bicarbonate, and elevated BAP concentrations, which was associated with an 11-fold increased risk of skeletal comorbidity (reduced standardized height, limb deformities, and/or requirement for orthopedic surgery of the lower extremities) compared to CKD controls [13]. The markedly elevated BAP levels in INC patients across all CKD stages suggest persistent mineralization defects in INC patients despite measures for Fanconi syndrome. INC patients showed a specific CKD stage-dependent pattern of bone markers (Figure 3). As expected in CKD controls, FGF23 and sclerostin levels started to increase as early as in CKD stage 2 (eGFR < 90 mL/min/1.73 m^2^). By contrast, INC patients demonstrated a delayed increase or lacked an increase in FGF23 and sclerostin serum levels in mild and moderate CKD [13]. FGF23 levels were independently associated with plasma phosphate, calcium and eGFR, suggesting that the delayed increase in FGF23 levels observed in INC patients compared to CKD controls was most likely due to concomitant hypophosphatemia and hypocalcemia. Both, eGFR and dosage of phosphate salts in INC patients correlated with sclerostin levels in the whole study cohort, suggesting that hypophosphatemia may prevent an increase in sclerostin levels in INC patients. Alternatively, the lack of increase in sclerostin levels may reflect a mechanism to compensate osteoblast malfunction due to *CNTS* mutations (vide supra). TRAP5b serum concentrations were elevated by approx. 1.7 SD score in INC patients compared to CKD controls, irrespective of eGFR. The diagnosis of INC was the only factor showing a significant association with sclerostin levels. TRAP5b is synthesized in osteoclasts and its serum levels were shown to correlate with osteoclast activity as well as numbers of osteoclasts. Therefore, elevated TRAP5b levels indicate that either the number of osteoclasts is increased or osteoblast activity is increased in INC patients, compared to other CKD patients of the same age and eGFR.

This confirms the results from the above-mentioned ex vivo studies, using PBMCs derived from INC patients, that *CTNS* malfunction results in increased osteoclastogenesis promoting increased bone resorption and reduced bone mass in INC patients. By contrast, no differences were observed with respect to OPG levels between INC patients and CKD controls [13].

Taken together, this study suggests that bone mineralization is markedly impaired in INC patients, despite treatment of Fanconi syndrome, which can only be partly normalized by kidney transplantation, and that CMBD is not only the consequence of Fanconi syndrome and progressive CKD resulting in CKD-MBD, but also an intrinsic osteoblast and osteoclast defect.

The distinct alterations in phosphate hemostasis and FGF23 levels in INC patients were later confirmed by Florenzano et al. [14]. They compared phosphate homeostasis and FGF23 levels in a cohort of 50 INC patients, with that of 97 CKD patients, matched for age and degree of renal insufficiency, with other underlying kidney diseases. INC patients displayed significantly lower circulating phosphate levels due to impaired tubular phosphate reabsorption and lower FGF23 concentrations compared to CKD controls, independent of eGFR. A multivariable analysis revealed that diagnosis of INC was independently associated with lower FGF23 levels after adjustment for age and eGFR. This study further supports the concept that phosphate is an important stimulator of FGF23 synthesis in bone and that the pathophysiology of bone disease in INC differs markedly from that in other CKD patients.

## 5. Towards an Interplay between Bone, Adipocytes, and Muscles

Osteoblasts, myocytes, and adipocytes both derive from mesenchymal stem cells (MSCs). As discussed above, myopathy is part of the complexity of CMBD, and the strength of load-bearing bones largely depends on growing muscle strength [40]. This is why the interplay between bone, adipocytes, and muscle is particularly relevant in patients with INC. We recently described the intrinsic cellular defects observed in bone cells from INC patients and animal models [11,12,49].

In 2016, Cheung et al. demonstrated the interplay between fat and muscle mass in INC in the murine *Ctns*^−/−^ model, showing profound muscle wasting, together with inhibited myogenesis, stimulated proteolysis, overexpressed pro-inflammatory cytokines (among them interleukin 1α, interleukin 6, and TNFα) in brown adipose tissue, hypermetabolism, upregulated thermogenesis, and increased presence of beige adipocytes [17]. The authors also describe the potential value of both native and active vitamin D in improving adipose tissue browning and muscle wasting in their murine *Ctns*^−/−^ model [51]. In this model, repletion of vitamin D improved (and sometimes normalized) weight gain, food intake, lean and fat mass; it also improved energy homeostasis as well as the size of skeletal muscle fibers and in vivo muscular function. From a molecular perspective, vitamin D repletion corrected abnormal expression of molecules, playing a key role in adipose tissue browning, and normalized the most important 20 differentially expressed genes in *Ctns*^−/−^ mice, as evaluated by muscle RNA-seq [51]. Whether vitamin D may exert similar effects in patients with INC remains to be proven. These results are, nevertheless, particularly relevant in the field of INC for two main reasons: 1. most patients with INC receive native vitamin D supplementation and sometimes active vitamin D analogs; and, 2. vitamin D is an anti-inflammatory agent [52].

Indeed, Prencipe et al., reported an upregulation of the IL1 β receptor 1 and IL1 receptor 2 in the kidneys of LPS treated animal models and in INC patients, this stimulation being driven by caspase 1 in the inflammasome [53]. It should be born in mind, that patients with INC displayed elevated levels of Il1β, Il18, and caspase 1 compared to healthy controls, but they also displayed significantly elevated levels of Il18 compared to patients with Familial Mediterranean Fever, a prototypical autoinflammatory disease [54]. Recently, Cheung et al., demonstrated that by treating *Ctns*^−/−^ mice with the Il1 receptor antagonist Anakinra they were able to attenuate the cachexia phenotype and correct the abnormal expression of the main biomarkers of beige adipose cells and adipocyte tissue browning, and to attenuate 12 of the 20 main, differentially expressed genes in *Ctns*^−/−^ mice [55]. Anakinra treatment also normalized muscle weight and fiber size and decreased infiltration in muscle fat [55]. These results were close to those observed earlier by the same team using vitamin D [51], questioning again the role of vitamin D in inflammation regulation.

Several pro inflammatory cytokines have been described as substitutes for RANKL, the factor synthesized by osteoblasts to promote osteoclastic differentiation, and, notably, interleukin 1β (IL1β) or interleukin 6 (IL6) [56]. If the interleukin-1 deregulation observed in cystinotic muscle is also proven to be true in other target tissues, notably in bone, this may open up a new avenue for therapeutic perspectives in INC.

## 6. Conclusions: Perspectives in Research

CMBD has emerged over the last decade as a well-recognized, long-term complication, inducing significant morbidity and impaired quality of life in teenagers and adults with INC. Its underlying pathophysiology is complex, consisting of abnormal mineral regulation, intrinsic bone defects, cysteamine toxicity, muscle wasting, and likely interleukin-1 driven inflammation. In future, biotherapies targeting the IL1β and/or Il6 pathway may improve muscle wasting and, subsequently, CMBD, but this remains to be proven. In the meantime, the promising results of stem cells transplantation on muscular cystine content in the murine model [57], should be confirmed in the currently ongoing clinical trials.

## Figures and Tables

**Figure 1 cells-11-00170-f001:**
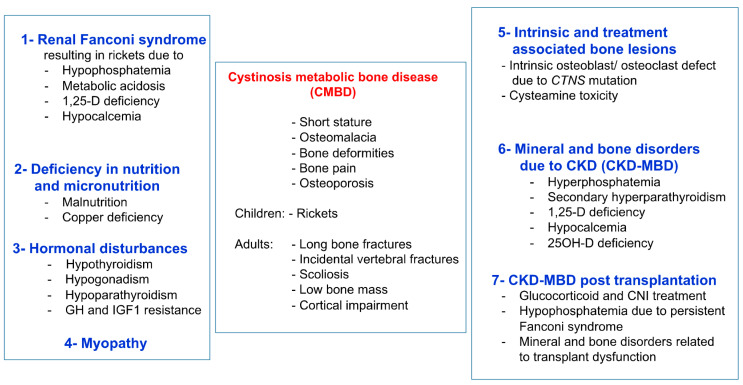
Clinical presentation and causes of CMBD. Figure from Hohenfellner et al., with permission [18].

**Figure 2 cells-11-00170-f002:**
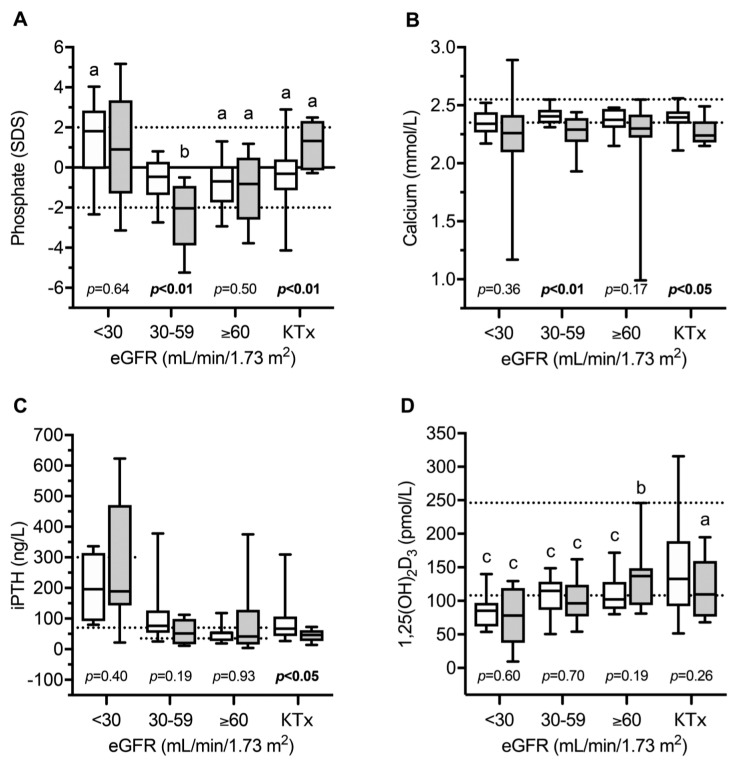
Serum levels of phosphate (**A**), calcium (**B**), intact parathyroid hormone (iPTH, (**C**)), and 1,25(OH)_2_D_3_ (**D**) in children with infantile nephropathic cystinosis (INC) and CKD controls as estimated glomerular filtration rate (eGFR) and after kidney transplantation (KTX). Gray box plots indicate INC patients; white box plots indicate CKD controls. Horizontal continuous and broken lines in (**A**) indicate the mean and upper and lower normal range; horizontal broken lines in (**B**,**C**) indicate the upper and lower normal range; horizontal broken lines in (**D**) indicate the PTH target range recommended by KDOQI; a, b, and c indicate *p* < 0.05, *p* < 0.01 and *p* < 0.001 versus healthy children, respectively. SDS, standard deviation score. Figure from Ewert et al., with permission [13].

**Figure 3 cells-11-00170-f003:**
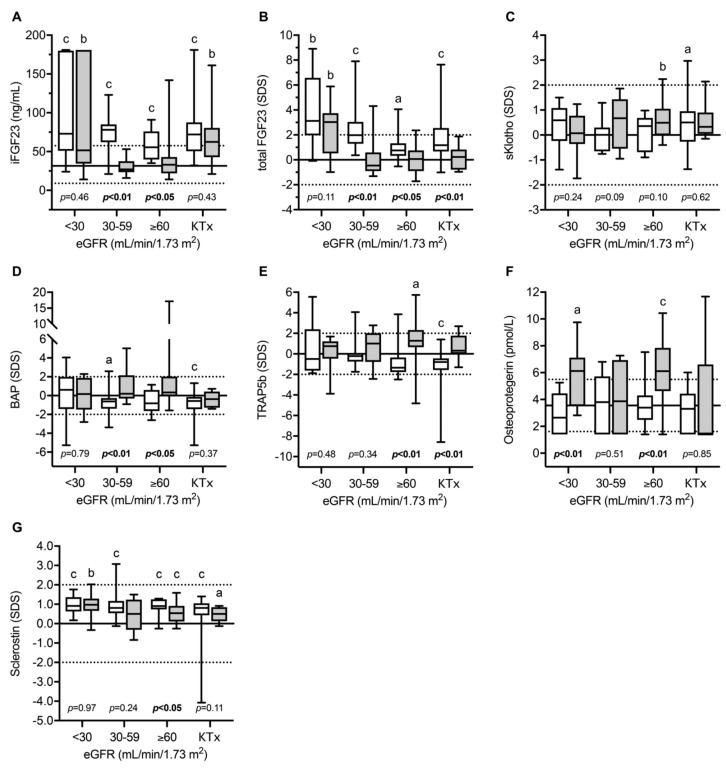
Circulating levels of intact (**A**) and total (**B**) fibroblast growth factor 23, soluble Klotho (**C**), bone alkaline phosphatase (**D**), tartrate-resistant acid phosphatase 5b (**E**), osteoprotegerin (**F**) and sclerostin (**G**) in children with infantile nephropathic cystinosis (INC), and CKD controls at various stages of CKD and after kidney transplantation (KTX): Gray box plots indicate INC patients while white box plots indicate CKD controls. Horizontal continuous and broken lines indicate the mean, upper, and lower normal range; a, b, and c indicate *p* < 0.05, *p* < 0.01, and *p* < 0.001 versus healthy children, respectively. eGFR, estimated glomerular filtration rate (eGFR); SDS, standard deviation score; iFGF23, intact fibroblast growth factor 23; BAP, bone alkaline phosphatase; TRAP5b, tartrate-resistant acid phosphatase 5b; OPG, osteoprotegerin; sKlotho, soluble Klotho. Figure from Ewert et al. with permission [13].

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
