# Peer review of "Muscle and Bone Impairment in Infantile Nephropathic Cystinosis: New Concepts"

_cells, 2022, doi:10.3390/cells11010170_

Round 1
Reviewer 1 Report
The authors present a detailed review of the current literature in relation to new concepts in muscle and bone impairment in infantile nephropathic cystinosis. The manuscript is of sufficient quality to be suitable for publication with the following minor revisions:
- Authors should include some discussion on the potential of the novel stem cell therapy in treatment of muscle and bone impairment in cystinotic patients.
- The review, while similar to a recent review (Machuca-Gayet et al 2021, IJMS), is sufficiently different to warrant publication. However inclusion of some discussion of the potential role of mitochondrial dysfunction should be considered.
Author Response
Authors should include some discussion on the potential of the novel stem cell therapy in treatment of muscle and bone impairment in cystinotic patients.
Reply: Thank you for this suggestion which we incorporated in the revised manuscript line 1412-14.
The review, while similar to a recent review (Machuca-Gayet et al 2021, IJMS), is sufficiently different to warrant publication. However inclusion of some discussion of the potential role of mitochondrial dysfunction should be considered.
Reply: Thank you for this suggestion which we incorporated in the revised manuscript line 952-956 .

Reviewer 2 Report
In this MS the authors have made a review of muscle and bone disease in nephropatic cysrinosis, a condition in which CKD patients have mineral abnormalities that are distinct from those in CKD stemming from other causes.The review is detailed and easy to follow. I have a few concerns, which are released to the authors as follows:
Major
-.. eventually copper deficiency which all contribute to impaired bone growth (25,26). the role of acidemia in impairing growth seems to be overlooked.
-Please clarify these two contradictory sentences:..line 165 increased expression in the number of sclerostin and FGF23-expressing osteocytes, causing progressive elevations in the plasma concentrations of these bone-derived factors
and, line 80:
introduction: .. However, there was a tendency towards low parathyroid hormone (PTH) and low fibroblast growth factor 23 (FGF23) levels..
-line 243-278. This is a very long description that could be usefully shortened.
-Line 315-319 Please consider that studies on vitamin D supplementation were essentially negative on muscle (Front Nutr. 2021 Aug 12;8:701386).
-line 250.. This extends the concept coming from studies in adult CKD patients that elevated circulating FGF23 and sclerostin are the earliest detectable changes of CKD-MBD to the pediatric CKD population.please add a reference
-Targeting IL1β pathway may improve muscle wasting..please consider that also IL6, which inot specific of the inflammasome seems to be a therapeutic target in CKD.
-the manuscripts needs english editing (assesment, 25 OH vitamin D, cysteramine,vitamin d, deformaties etc..)
Minor
Figure permissions need to be enclosed
Author Response
-.. eventually copper deficiency which all contribute to impaired bone growth (25,26). the role of acidemia in impairing growth seems to be overlooked.
Reply: This important topic was incorporated in the revised manuscript on line 180-181.
-Please clarify these two contradictory sentences:..line 165 increased expression in the number of sclerostin and FGF23-expressing osteocytes, causing progressive elevations in the plasma concentrations of these bone-derived factors
and, line 80:
introduction: .. However, there was a tendency towards low parathyroid hormone (PTH) and low fibroblast growth factor 23 (FGF23) levels..
Reply: Sorry for this misleading sentences. We now make clear in the revised manuscript that in CKD patients suffering from other causes than cystinosis the first sign of bone disease is an increased secretion of FGF23 and sclerostin in bone, whereas this is not the case in cystinosis.
-line 243-278. This is a very long description that could be usefully shortened.
Reply: This section was substantially shortened.
-Line 315-319 Please consider that studies on vitamin D supplementation were essentially negative on muscle (Front Nutr. 2021 Aug 12;8:701386).
Reply: This was edited accordingly.
-line 250.. This extends the concept coming from studies in adult CKD patients that elevated circulating FGF23 and sclerostin are the earliest detectable changes of CKD-MBD to the pediatric CKD population.please add a reference
Reply: This sentence was deleted in order to shorten this section.
-Targeting IL1β pathway may improve muscle wasting..please consider that also IL6, which inot specific of the inflammasome seems to be a therapeutic target in CKD.
Reply: This was edited accordingly.
-the manuscripts needs english editing (assesment, 25 OH vitamin D, cysteramine,vitamin d, deformaties etc..)
Reply: The manuscript was edited by a native english speaker (Angela Bain Emslie) and all changes are given in the tracking mode.
Minor
Figure permissions need to be enclosed
Reply: Last time this was not possible for me to upload this, just could not find the right button,. hope that this will be possible now.
Round 2
Reviewer 2 Report
Thank you for nice review